# Curing Behavior, Rheological, and Thermal Properties of DGEBA Modified with Synthesized BPA/PEG Hyperbranched Epoxy after Their Photo-Initiated Cationic Polymerization

**DOI:** 10.3390/polym12102240

**Published:** 2020-09-29

**Authors:** Tossapol Boonlert-uthai, Kentaro Taki, Anongnat Somwangthanaroj

**Affiliations:** 1Department of Chemical Engineering, Faculty of Engineering, Chulalongkorn University, Bangkok 10330, Thailand; tossapol.bu@gmail.com; 2School of Mechanical Engineering, Kanazawa University, Kanazawa 920-1192, Japan; taki@se.kanazawa-u.ac.jp

**Keywords:** hyperbranched epoxy, photo-polymerization, rheological property, thermal property

## Abstract

This paper investigates the photo-initiated cationic polymerization of diglycidyl ether of bisphenol A (DGEBA) modified with bisphenol A (BPA)/polyethylene glycol (PEG) hyperbranched epoxy resin. The relationship between curing behavior, rheological, and thermal properties of the modified DGEBA is investigated using photo-differential scanning calorimetry (DSC) and photo-rheometer techniques. It is seen that the addition of the hyperbranched epoxy resin can increase UV conversion (α_UV_) and reduce gelation time (*t*_gel_). After photo-initiation polymerization (dark reaction) occurred, a second exothermic peak in the DSC thermogram takes place: namely, the occurrence of curing reaction owing to the activated monomer (AM) mechanism. Consequently, the glass transition temperature decreased, and at the same time, UV intensity increased which was due to the molecular weight between crosslinking points (*M*_c_). Furthermore, the radius of gyration (*R*_g_) of the network segment is determined via small-angle X-ray scattering (SAXS). It is noted that the higher the *M*_c_, the larger the radius of gyration proves to be, resulting in low glass transition temperature.

## 1. Introduction

In the electronics, coatings, and automotive industries, UV curing application is extensively used. This is because of the many advantages of UV curable resins, such as high reaction rate, good adhesion, curing at ambient temperature, and low energy consumption [1,2]. In addition, this single component, solvent free type of resin has high stability at any storage condition. Photo-initiated cationic polymerization is a curing reaction of the thermosetting materials, especially epoxy systems. Once a photoinitiator absorbs UV light, the photoinitiator is cleaved by UV light and changed into radicals and protonic acids or Brönsted acids. They can initiate cross-linking polymerization with a monomer. Polymer networks are finally formed. The photoinitiator used for the photo-initiated cationic polymerization is mostly onium salts, such as diaryliodonium salt and triarylsulfonium salt [3]. Crivello [3], one of the best photo-polymerization experts, proposed the overall mechanism of photolysis of a diaryliodonium salt $\left({\mathrm{Ar}}_{2}I^{+}{\mathrm{MtX}}_{n}^{-}\right)$ and a monomer (M), as described in Equations (1)–(4):(1)$${{\mathrm{Ar}}_{2}I}^{+}{\mathrm{MtXn}}^{-}\overset{h\upsilon }{⟶}\left[\begin{array}{l} {\mathrm{ArI}}_{\cdot }{}^{+}{\mathrm{MtXn}}^{-}+\mathrm{Ar}\cdot \\ {\mathrm{Ar}}^{+}{\mathrm{MtXn}}^{-}+\mathrm{ArI} \end{array}\right]$$
(2)$$\left[\begin{array}{l} {\mathrm{ArI}}_{\cdot }{}^{+}{\mathrm{MtXn}}^{-}+\mathrm{Ar}\cdot \\ {\mathrm{Ar}}^{+}{\mathrm{MtXn}}^{-}+\mathrm{ArI} \end{array}\right]\overset{\mathrm{solvent}\;\mathrm{or}}{\underset{\mathrm{monomer}}{⟶}}{\mathrm{HMtX}}_{n}$$
(3)$${\mathrm{HMtX}}_{n}+M⟶{\mathrm{HM}}^{+}{\mathrm{MtX}}_{n}{}^{-}$$
(4)$${\mathrm{HM}}^{+}{\mathrm{MtX}}_{n}{}^{-}+nM⟶{H(M)}_{n-1}M^{+}{\mathrm{MtX}}_{n}{}^{-}$$

The photoinitiator bearing anions viz., ${\mathrm{MtX}}_{n}^{-}$ affects the polymerization rate of photoinitiated cationic polymerization. The order of ${\mathrm{MtX}}_{n}^{-}$ increasing polymerization rates is as follows: ${\mathrm{SbF}}_{6}^{-}>{\left(C_{6}F_{6}\right)}_{4}B^{-}>{\mathrm{AsF}}_{6}^{-}>{\mathrm{PF}}_{6}^{-}>{\mathrm{BF}}_{6}^{-}>{\mathrm{ClO}}_{4}^{-}$ [1,3,4,5]. In the presence of alcohol, the ring-opening cationic polymerization of epoxy takes place with two different mechanisms: the activated chain end (ACE) and the activated monomer (AM) mechanisms, as shown in Figure 1 [6]. The ACE mechanism is a general reaction for the ring cationic polymerization of the epoxy system. First, the acid initiator reacts with an epoxy monomer. Then, there is the formation of an activated monomer or propagating center. Next, the activated monomer reacts with monoalcohol by transfer reaction, resulting in the formation of a dead end. Concurrently, the dead-end (OH group) continues to react with the activated monomer to produce a protonated ether and forms a new hydroxyl-terminated extended molecule by deprotonation by the epoxide monomer. As a result, the growing polymer chain was terminated but it can continue to react with the activated monomer. Repetition of this mechanism regenerates a hydroxyl-terminated chain that continues the termination and transfers reactions. Therefore, each time the OH group reacts, another OH group is produced; and this condition is called the AM mechanism. The rates of ACE and AM mechanisms [7] are shown in Equations (5) and (6), respectively:
R_ACE_ = k_ACE_[M^+^][M](5)
R_AM_ = k_AM_[M^+^][ROH](6)
where k_AEC_ is the rate constant of ACE mechanism, k_AM_ is the rate constant of AM mechanism, [M] is a concentration of monomer M, [M^+^] is a concentration of activated monomer M^+^, and [ROH] is a concentration of alcohol ROH.

In a complex area or large thickness of a sample, UV curing reaction may be incomplete because UV light cannot penetrate into deep layers [8]. During curing reaction, the glass transition temperature (*T*_g_) of a sample increases [9] until it forms a gel. If the temperature is not higher than the glass transition temperature at the gel point (*T*_g, gel_), vitrification occurs, resulting in the restriction of the curing reaction [1,10]. Therefore, the reaction has to be performed at a suitable temperature: at a higher temperature. Higher UV intensity also needs to be applied [11] in order to accelerate the reaction rate and to complete the curing reaction [12,13].

The commercial epoxy thermoset DGEBA is widely used because it has high thermal and mechanical properties, high chemical resistance, and low shrinkage. Yet, DGEBA is known to have some inherent adverse properties: brittleness and low toughness [1,14,15]. Nevertheless, a number of researchers have been engaged in developing properties that have toughening agents or plasticizers, especially PEG, to improve the low impact resistance of DGEBA by decreasing *T*_g_ [16,17,18,19]. Researchers have also studied the mechanisms that affect the final properties of the epoxy thermoset.

In recent years, hyperbranched polymers have been used for industrial-scale productions and applications such as additives and resins for high-performance materials, rheology modifiers, diluents, and crosslinkers [20]. This is because hyperbranched polymers are easy to synthesize, and have low viscosity, high solubility, as well as a large number of desired end functional groups. In our previous research [21], the hyperbranched epoxy resins, consisting of BPA and PEG reactants together with pentaerythritol (PE) as a branching point, have been successfully synthesized and characterized, as shown in Figure 2. Hyperbranched epoxy resins have gained strength from the BPA structure and flexibility from the PEG structure. Thermally cured hyperbranched epoxy is highly resistant and stable because it does not have a melting point that is the defect of the thermosetting materials [19,22].

This research aims to improve UV curing reaction and the inherent adverse properties of DGEBA. The study focuses on the relationship of curing behavior, rheological, and thermal properties of DGEBA modified with hyperbranched epoxy resin through photo-initiated cationic polymerization. The curing reaction and thermal properties have been investigated via photo-DSC and are seen to relate to the rheological properties using a photo-rheometer. The radius of gyration of the network segment has been evaluated using the small-angle X-ray scattering (SAXS) technique to examine the relationship among them.

## 2. Materials and Methods

### 2.1. Materials

DGEBA, having an epoxy equivalent weight (EEW) of 170.21 g eq^−1^, (Sigma-Aldrich, St. Louis, MO, USA) was used as the main base of epoxy resin in the formulated mixture. Hyperbranched epoxy resin was synthesized and characterized via FTIR and NMR techniques. The results are shown in Appendix A, respectively. The chemical for the synthesis: BPA (Tokyo Chemical Industry Co., Ltd., Tokyo, Japan) was used as an aromatic monomer, and was purified via recrystallization using toluene before use. PEG400, Mw = 400 g/mol, (Sigma-Aldrich, St. Louis, MO, USA) was used as an aliphatic monomer. Epichlorohydrin (ECH), (Tokyo Chemical Industry Co., Ltd., Tokyo, Japan) was used as an epoxidation reagent. PE (Tokyo Chemical Industry Co., Ltd., Tokyo, Japan) was used as a branch generating unit and was recrystallized with ethanol before use. Sodium hydroxide (NaOH) as a base catalyst and sodium chloride (NaCl) were obtained from Ajax Finechem, Sydney, Australia. Hydrobromic acid (HBr), acetic acid, potassium acid phthalate, methyl violet, and chlorobenzene (Tokyo Chemical Industry Co., Ltd., Tokyo, Japan) were used to determine the epoxy equivalent weight (EEW) following the standard test methods (ASTM D 1652) [23]. Triarylsulfonium hexafluorophosphate salts (Sigma-Aldrich, St. Louis, MO, USA) were used as a photoinitiator.

### 2.2. Synthesis of Hyperbranched Epoxy Resin

The hyperbranched epoxy resins were synthesized by the A_2_ + B_4_ polycondensation reactions [6] using PEG400 and BPA as A_2_ monomers and PE as B_4_ monomer. The amount of B_4_ monomer was 10 wt % of A_2_ monomer’s content and the mass ratio of BPA to PEG400 was 90:10 (HBE10P). The molar ratio of ECH to A_2_ monomers was fixed at 2:1. First, PE, BPA, and ECH were mixed in a two-necked RB flask installed with a condenser and dropping funnel at 60 °C. 5N aqueous NaOH solution was slowly dripped into the mixture for 30 min and the reaction temperature was set at 110 °C for 4 h when the desired time was completed, the reaction was terminated by immediately quenching the mixture. The mixture was then poured into a separation funnel in order to separate the aqueous layer (the residual reactants) from the organic layer. The organic layer was purified by washing with 15 wt % NaCl solution followed by distilled water until the washer’s pH was 8–9. Finally, the organic solution was dried in a vacuum oven at 70 °C until the mass of the sample was constant and viscous transparent liquid was observed.

### 2.3. Preparation of Epoxy Mixtures

There are two systems of epoxy mixtures: the DGEBA system (100 wt % of DGEBA) and the D90H10 system (90 wt % of DGEBA and 10 wt % of HBE10P, EEW = 564 g eq^−1^). The weight of both resins was 10 g. As for the D90H10 system, DGEBA and synthesized HBE10P resins were homogenously blended for 10 min at 40 °C. After blending, 0.5 g triarylsulfonium hexafluorophosphate salts (5 wt % of resin) were added to the resins and the mixture was mechanically stirred at room temperature for 10 min. In Table 1, the code for curing condition is IxxTyytzz where *I* is UV intensity (xx = 10, 20, 30, 40 and 50 mW/cm^2^), *T* is temperature (yy = 80 °C) and *t* is irradiation time (zz = 60 s). For example, *I*10*T*80*t*60 code is the condition using UV intensity of 10 mW/cm^2^, *T* = 80 °C, and *t* = 60 s.

The mixture EEW was calculated, as expressed in Equation (7):(7)$$\mathrm{EEW}\;\mathrm{mixture}=\frac{\mathrm{Total}\;\mathrm{Wt}}{\frac{{\mathrm{Wt}}_{a}}{{\mathrm{EEW}}_{a}}+\frac{{\mathrm{Wt}}_{b}}{{\mathrm{EEW}}_{b}}}$$

### 2.4. Curing Behavior and Thermal Property

Both curing behavior and thermal properties of the epoxy systems were investigated via photo-DSC (PerkinElmer: DSC8500, Waltham, MA, USA), as shown in Appendix A, under nitrogen atmosphere: three times for each system. The UV source and high-pressure mercury lamp were from Omicure Series2000 (Excelitas Technologies, Waltham, MA, USA). For the purpose of calibrating UV intensity, a carbon black plate was used in the sample cell. The intensity was determined through the relationship between the power of UV light and UV energy. This UV energy is generated through the absorption of UV light by the carbon black plate. The uncured samples (8.0 ± 0.2 mg) were placed in aluminum pans having quartz covers and put under isothermal condition for 1 min before and 3 min after UV exposure. For the first scanning, UV curing was performed at *T* = 80 °C for 1 min followed by UV irradiation of 10, 20, 30, 40, and 50 mW/cm^2^ for 1 min. After UV irradiation, the dark reaction proceeded under isothermal condition (80 °C) for 3 min. The residual heat of reaction of the cured samples was measured via a second scan in the range of 30–180 °C, along with a heating rate of 10 °C/min to inspect the heat profile. Lastly, the second protocol was repeated to determine the glass transition temperature (*T*_g_) of the cured samples. Conversion (α) or degree of cure of the cured samples was evaluated, as in Equation (8):(8)$$\alpha =\frac{\Delta H_{1}}{\Delta H_{\mathrm{total}}};\;\Delta H_{\mathrm{total}}=\Delta H_{1}+\Delta H_{\mathrm{res}}$$
where $\Delta H_{\mathrm{total}}$ is the total heat of reaction, $\Delta H_{1}$ is the heat of reaction in the first scan, and $\Delta H_{\mathrm{res}}$ is the residual heat of reaction in the second scan.

### 2.5. Rheological Property

The rheological property of the epoxy systems was determined by the photo-rheometer MCR-302WEPS (Anton-Paar, Graz, Austria), as shown in Appendix A. The UV source and high-pressure mercury lamp were from Omicure Series2000 (Excelitas Technologies). The rheological properties of the uncured samples were evaluated within the linear viscoelastic region using a parallel plate (PP12-Dispo, d = 12 mm) with a gap between parallel plates of 0.1 mm, frequency of 10 Hz, strain of 0.1%, and isothermal temperature of 80 °C. Each system has been evaluated three times to confirm the repeatability of results. To maintain initial viscosity and initial modulus of the uncured samples, they were held under isothermal condition for 1 min before the curing reaction took place, as shown in Figure 3. The UV curing experiments were performed applying five UV intensities i.e., 10, 20, 30, 40, and 50 mW/cm^2^ for 1 min.

### 2.6. Radius of Gyration of the Network Segment

The radius of gyration was investigated via small-angle X-ray scattering (SAXS) technique (Nano Viewer RA-MICRO7HFM, Rigaku, Tokyo, Japan). The wavelength of the incident X-ray (CuKα) was 0.154 nm and the camera length, the sample-to-detector distance, was 700 mm. The range of the scattering vector length (q) was 0.014–0.355 nm^−1^. The cured samples, DGEBA and D90H10 systems, obtained from photo-DSC experiments, were measured under isothermal condition (25 °C). The size of the imaging plate (IP) detector was 125 × 125 mm.

The radius of gyration (*R*_g_) indicates a nanodomain dispersed in the epoxy matrix, as shown in Figure 4 and was evaluated by the Zimm plot (1/I vs q^2^). The Ornstein-Zernike model [24] was assumed for the q-dependence of the scattering intensity *I*(q), as in Equations (9) and (10):(9)$$I\left(q\right)=\frac{I_{0}}{1+\xi^{2}q^{2}}$$
(10)$$\frac{1}{I}=\frac{1}{I_{0}}+\frac{\xi^{2}q^{2}}{I_{0}}$$
where ξ is the correlation length and *I*_0_ is the absolute intensity. A plot of 1/*I* vs. q^2^ (Equation (10)) produces 1/*I*_0_ (intercept) and ξ^2^/*I*_0_ (slope). At low q, the radius of gyration was evaluated, as shown in Equation (11):(11)$$R_{g}=\sqrt{3}\xi$$

### 2.7. Molecular Weight Between Crosslinking Points

The molecular weight between crosslinking points (*M*_c_) is simply the average molecular weight of the monomer (*M*_av_) divided by the number of cross-links per molecule (c), as in Equation (12):(12)$$M_{c}=\frac{M_{av}}{c}$$

Even though it is difficult to evaluate the number of cross-links per molecule, there is a relationship between the network structure and gelation [25]. The modulus of the entangled polymer network can be approximated, as a simple sum, as shown in Equation (13):(13)$$G≅G_{x}+G_{e}\approx \rho \mathrm{RT}\left(\frac{1}{M_{x}}+\frac{1}{M_{e}}\right)$$
where G_x_ is the modulus of all classic models, G_e_ is the rubbery plateau modulus of high molar mass polymer, ρ is the density of the cured sample, R is the gas constant (8.314 m^3^Pa K^−1^ mol^−1^), *T* is temperature (K), *M*_x_ is the apparent molar mass, and *M*_e_ is the entanglement molar mass. At the gel point, entanglement effects become negligible when compared with the effects of covalent cross-links, such as high viscosity and high storage modulus; therefore, the ${G^{\prime }}_{\mathrm{gel}}$ can be expressed, as in Equation (14):(14)$${G^{\prime }}_{\mathrm{gel}}\approx \frac{\rho \mathrm{RT}}{M_{c}}$$
where ${G^{\prime }}_{\mathrm{gel}}$ is the storage modulus at the gel point and the ρ of each sample was determined by the density kit MS-DNY-54 (Mettler Toledo, Greifensee, Switzerland), as presented in Appendix A.

## 3. Results and Discussion

### 3.1. Curing Behavior and Rheological Property at Various Photo-Curing Conditions

The curing behavior and rheological property of the DGEBA and D90H10 systems, using triarylsulfonium hexafluorophosphate salts acting as photoinitiator, were investigated via photo-DSC and photo-rheometer, respectively. These properties were studied by varying UV intensity (10, 20, 30, 40, and 50 mW cm^−2^), having an irradiation time of 60 s and temperature of 80 °C. Photocuring was performed at 80 °C to avoid vitrification during the reaction and to follow the real process of the hard disk drive production wherein photocuring was performed at high temperature.

In terms of the rheological property of epoxy resin, an important parameter is gelation time (*t*_gel_). Gelation time is the time of the curing reaction for all monomers approaching maximum molecular weight between crosslinking points. As shown in Appendix A, the *t*_gel_ can be found at the crossover of storage modulus (G′) and loss modulus (G″): gelation point. As tabulated in Table 2, it is clear that when UV intensity increased, the *t*_gel_ of both systems decreased because of the high concentration of the activated photoinitiator, generating a high reaction rate. As shown in Figure 5a,b, the photo-rheological properties were seen to be complementary to the UV conversion ($\alpha_{\mathrm{UV}}$) from DSC’s heating profile. Thus, higher UV conversion of epoxy with low *t*_gel_ was observed and the increase in UV intensity was found to accelerate curing reaction. However, the conversion and cure rate of the D90H10 system proved to be higher than that of the DGEBA system because of the ball-bearing effect [26,27,28] and the branching of the hyperbranched structure, resulting in more mobility and more probability of reaction, as illustrated in Figure 6. Because of the increase and influence of the OH group in the system, there was a greater increase in the AM mechanism.

After being exposed to UV curing, the behavior of the DGEBA and D90H10 systems were not the same. This study used the molecular weight between crosslinking points (*M*_c_) in order to explain the curing reaction, which was calculated, as in Equation (14). As shown in Table 2, with regards to the DGEBA system, when UV intensity increased, *M*_c_ decreased. This decrease occurred because of the activated photoinitiator molecules, which gave rise to more protonated epoxide molecules. It was found, therefore, that the protonated epoxide chains reacted too fast, resulting in short-chain or low M_c_, as illustrated in Figure 7. As shown in Figure 5c, considering the cure rate (dα/dt), there were double peaks of the cure rate profile during the UV curing. The second peak dominated the first peak when UV intensity was more than 30 mW/cm^2^. The first peak depicts the formation of a network structure at the gel point; the second peak depicts the reaction of the activated molecules trapped in the network structure having high storage modulus at gelation time (G′_gel_), as shown in Table 2. When the systems were exposed to UV intensity of 10–30 mW/cm^2^, the first peak was observed. However, when UV intensity increased from 30–50 mW/cm^2^, the second peak predominated over the first peak. This result was due to the high concentration of the activated photoinitiator, which accelerated the UV curing reaction.

As regards the D90H10 system, the curing behavior and other properties were different from those of the DGEBA system. When UV intensity was raised to 30 mW/cm^2^, *M*_c_ increased because of the hyperbranched structure and high molecular weight of the HBE10P resin. The increase in activated photoinitiator molecules and OH groups promoted more protonated epoxide molecules in HBE10P and more long chains because of the AM mechanism which in turn created large molecules between the crosslinking points. However, this occurrence was reversed when UV intensity increased from 30 to 50 mW/cm^2^; thus, molecular weight between the crosslinking points greatly decreased. When excessive UV intensity is applied, the curing reaction is dependent on the DGEBA resin because the reactivity of DGEBA resin is much higher than that of the HBE10P resin whose UV conversion proved to be extremely low. Besides, it was noted that the excessive amount of activated photoinitiator molecules and protonated epoxide molecules resulted in very low *M*_c_. It was also clear that the cure rate slightly decreased and gelation time increased. This increase can be denoted by the high storage modulus at the gel point (G′_gel_), which induced low mobility and low cure rate during the UV reactions. Finally, the D90H10 system proved to have a higher number of cross-links per molecule (c) than the DGEBA system. This outcome was due to the presence of the dangling HBE10P chains in the network structure which can react when in contact with an activated monomer, as depicted in Figure 8.

As shown in Figure 5c,d, it is seen that there were double peaks in the cure rate profile of both systems: UV and dark curing. The first peak is the UV curing and the second peak is the dark curing (after shutting UV light off). Because of the presence of hydroxyl group in the DGEBA and HBE10P resins, such phenomena can be explained in that there are two propagation mechanisms of the ring-opening cationic polymerization of epoxy in the presence of the hydroxyl group: ACE and AM mechanisms [6]. During UV reaction, the ACE mechanism predominates until the reaction surpasses the maximum rate. Subsequently, the AM mechanism predominates because of the concentration of the hydroxyl group, which is larger than that of the initiator [*I*^+^]. After shutting off UV intensity (no generation of initiator), dark reaction follows the AM mechanism which is the chain transfer reaction, resulting in an activated monomer and the consumption of the OH group to produce another OH group. These reactions release heat. Thereby, the exothermic heat was observed via photo-DSC, culminating in a peak in the dark curing period. Moreover, the second peak at high UV intensity was seen to be higher than the second peak at low UV intensity, because of greater initiator concentration.

### 3.2. The Thermal Properties and Radius of Gyration of the Cured Samples at Various Curing Conditions

Glass transition temperature (*T*_g_) is usually used as a parameter to identify thermal stability during use in any application. For example, the operating temperature in a hard disk drive (HDD) is between 25 and 80 °C. Therefore, *T*_g_ for any part in the HDD must not be in the range of the operating temperature. In Figure 9 and Table 2, the glass transition temperature for all samples is tabulated. When UV intensity increased, the *T*_g_ of the DGEBA and D90H10 systems decreased. However, the *T*_g_ of the D90H10 system was found to be lower than that of the DGEBA system because of the internal plasticizer effect of PEG’s structure in the HBE10P resin, in which there were ether groups along with the free volume of the hyperbranched structure [21,29,30,31,32,33]. As for the epoxy system, the glass transition temperature generally increases when UV intensity, irradiation time, and temperature increase [34,35]. Besides, both the radius of gyration (*R*_g_) and the distance from the center of the network section in the samples were investigated in order to describe and confirm this occurrence. In Appendix A, the light scattering profiles of all samples measured via SAXS are shown. Subsequently, the *R*_g_ of the network segment was evaluated by the Zimm plot (1/*I* vs. q^2^), as depicted in Figure 10. In Appendix A, the results of the calculation are tabulated.

As shown in Table 2, a relationship was found between *R*_g_ and *M*_c_, for each sample. As a result, when *M*_c_ increased for both the DGEBA and D90H10 systems, *R*_g_ increased. As regards the DGEBA system, when UV intensity increased, *R*_g_ decreased. The D90H10 system, however, did not follow the same trend because of the addition of the HBE10P resin. Again, when UV intensity increased from 10 to 30 mW/cm^2^, *R*_g_ increased because there were more protonated epoxide molecules in HBE10P, and it could react with others to form large molecules between crosslinking points, which resulted in low cross-link density and low *T*_g_. Yet, when UV intensity increased from 30 to 50 mW/cm^2^, *R*_g_ decreased because *M*_c_ greatly decreased. There were also many activated photoinitiator molecules and protonated epoxide molecules, generating a lot of active chains. Each active chain quickly encountered each other, and this resulted in very low *M*_c_, high cross-link density and high *T*_g_. Because of predomination of the internal plasticizer effect from PEG’s structure in the HBE10P resin [21], *T*_g_ increased slightly.

## 4. Conclusions

In this work, photo-DSC, photo-rheometer and SAXS measurements were used to investigate the occurrence of the epoxy systems during UV curing and the final properties of the cured samples. Through the relationship between curing behavior, rheological and thermal properties, the short gelation time led to an increase in UV conversion and cure rate. Moreover, the values of the molecular weight between the crosslinking points, radius of gyration of the network segment, and glass transition temperature were seen to be congruent. It was noted that the addition 10 wt % of HBE10P affected the characteristics of the DGEBA system. Thus, the ball-bearing effect and branching structure of the hyperbranched epoxy increased the mobility of the system resulting in the improvement of conversion and cure rate. As *T*_g_ decreased, the low impact resistance of the DGEBA thermoset was enhanced accordingly. After UV irradiation in the DSC thermogram, the AM mechanism exhibited a second exothermic peak. In conclusion, the glass transition temperature for both the DGEBA and D90H10 systems depends on the reaction mechanism, the structure of resins, and UV intensity.

## Figures and Tables

**Figure 1 polymers-12-02240-f001:**
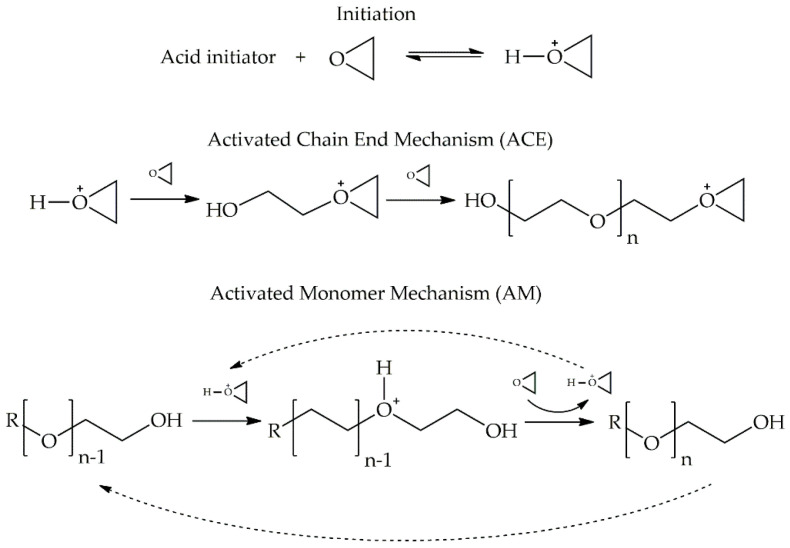
The mechanisms of the ring-opening cationic polymerization of epoxy in the presence of alcohol: the activated chain end (ACE) and the activated monomer (AM) mechanisms [6].

**Figure 2 polymers-12-02240-f002:**
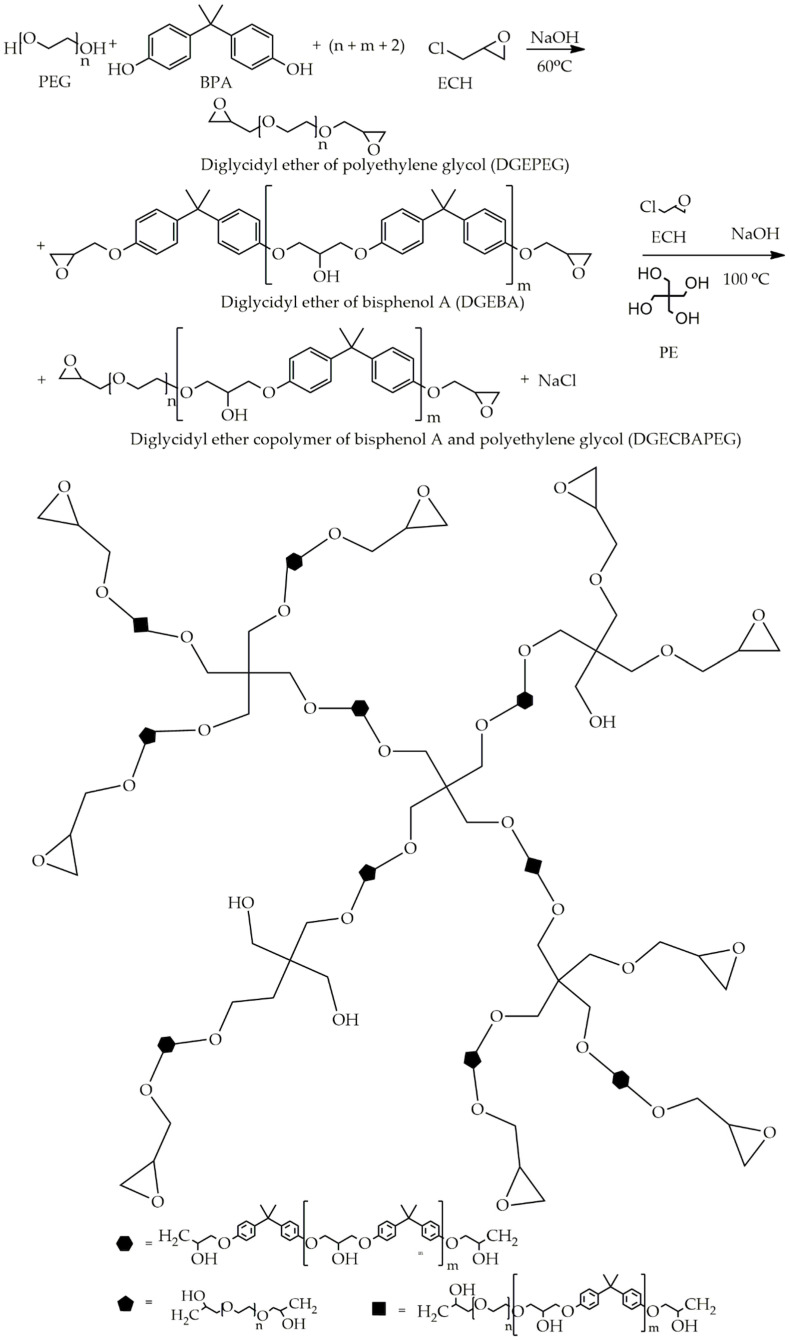
The synthesis and possible structure of bisphenol A/polyethylene glycol (BPA/PEG) hyperbranched epoxy resin [21].

**Figure 3 polymers-12-02240-f003:**
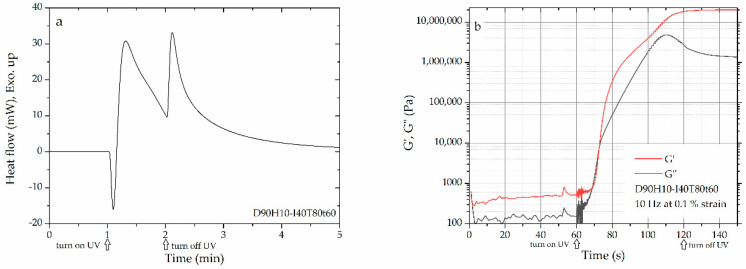
The representative data of (**a**) differential scanning calorimetry (DSC) and (**b**) rheological measurements.

**Figure 4 polymers-12-02240-f004:**
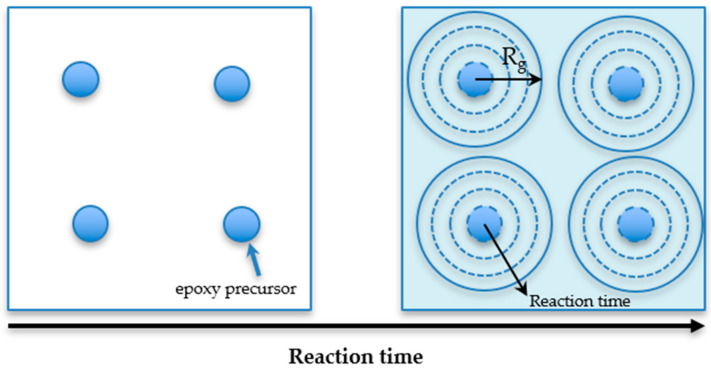
The transition of epoxy nanodomains through reaction time.

**Figure 5 polymers-12-02240-f005:**
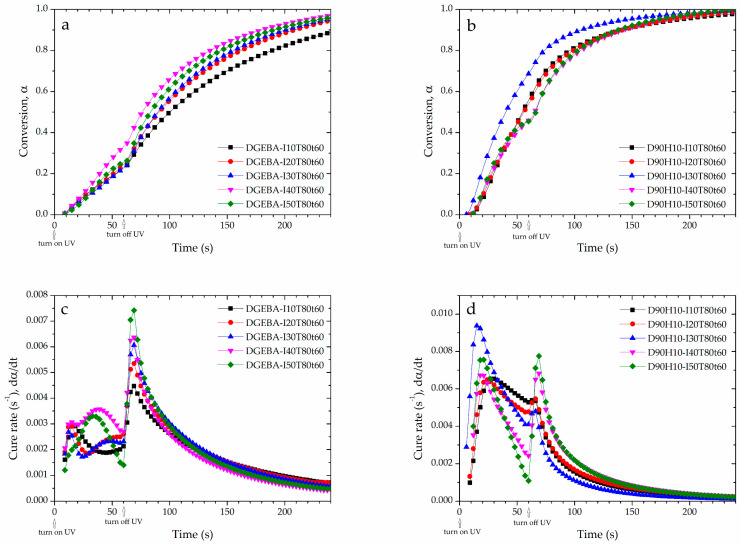
Conversion of (**a**) DGEBA and (**b**) D90H10, and cure rate of (**c**) DGEBA and (**d**) D90H10 with various UV intensity, the temperature of 80 °C and irradiation time of 60 s.

**Figure 6 polymers-12-02240-f006:**
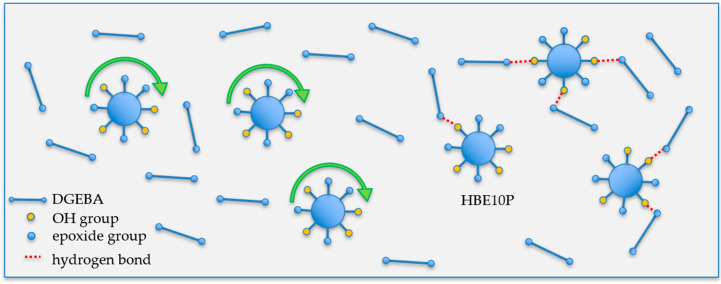
The effects of ball-bearing, globular and non-entanglement structure, and hydrogen bonds on the epoxy system.

**Figure 7 polymers-12-02240-f007:**
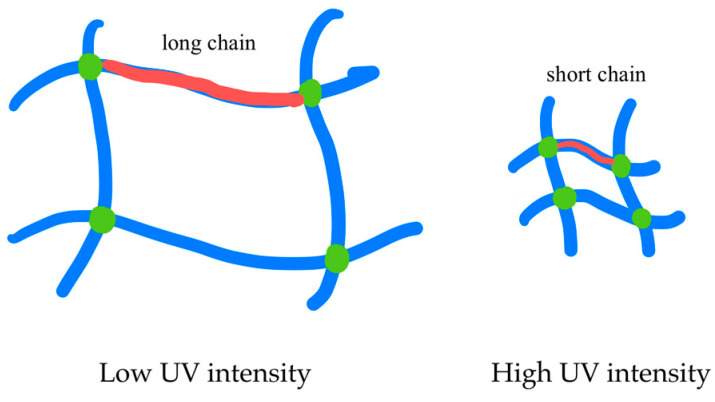
Effect of UV intensity on formation of network structure through photo-cationic polymerization.

**Figure 8 polymers-12-02240-f008:**
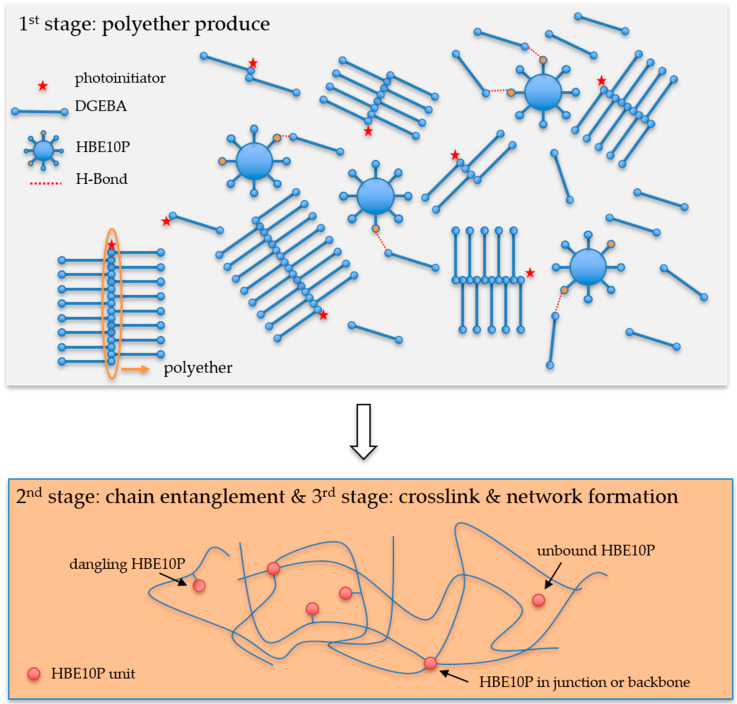
Scheme of UV curing reaction for DGEBA and HBE10P resins.

**Figure 9 polymers-12-02240-f009:**
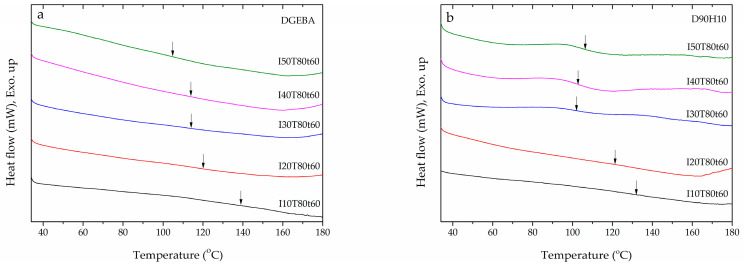
DSC thermograms showing glass transition temperature of (**a**) DGEBA and (**b**) D90H10 thermosets cured at various UV intensity.

**Figure 10 polymers-12-02240-f010:**
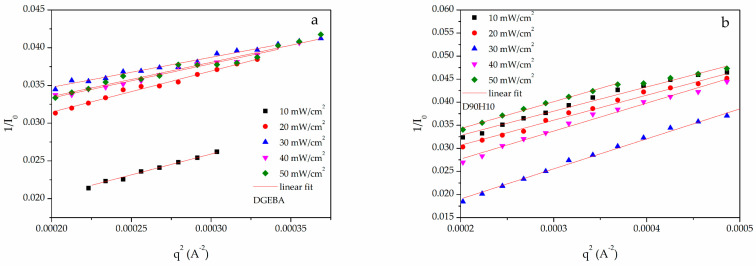
Zimm plot for (**a**) DGEBA and (**b**) D90H10 thermosets cured at various UV intensity.

**Table 1 polymers-12-02240-t001:** The mixture of epoxy systems.

System	EEW (g eq^−1^)	DGEBA Resin		HBE10P Resin		Photoinitiator
		wt %	Mass (g)	wt %	Mass (g)	Mass (g)
**DGEBA**	170.21	100	10.0	-	-	0.5
**D90H10**	182.98	90	9.0	10	1.0	0.5

**Table 2 polymers-12-02240-t002:** The properties of DGEBA and D90H10 systems at various UV intensity.

UV Intensity (mW/cm^2^)	Photo-Rheometer				Photo DSC			SAXS
	t_gel_ (s)	$G_{gel}^{\prime }$ (Pa)	M_c_ (g mol^−1^)	Number of Cross-Links Per Molecule	$\alpha_{UV}$	α	T_g_ (°C)	R_g_ (nm)
**DGEBA**								
10	41.0 ± 0.3	2410 ± 10	1,028,532 ± 245	0.0003 ± 1 × 10^−6^	0.23 ± 0.01	0.90 ± 0.01	140.15 ± 0.35	13.42 ± 0.14
20	31.6 ± 0.3	5025 ± 30	492,469 ± 141	0.0007 ± 3 × 10^−6^	0.25 ± 0.01	0.95 ± 0.01	120.72 ± 0.57	8.90 ± 0.00
30	23.8 ± 0.3	4080 ± 30	608,043 ± 345	0.0006 ± 4 × 10^−6^	0.23 ± 0.01	0.95 ± 0.00	114.22 ± 0.32	6.67 ± 0.01
40	17.7 ± 0.3	3890 ± 10	636,658 ± 940	0.0005 ± 1 × 10^−6^	0.32 ± 0.01	0.97 ± 0.01	114.33 ± 0.26	7.51 ± 0.04
50	14.5 ± 0.3	3412 ± 30	727,820 ± 8110	0.0005 ± 5 × 10^−6^	0.25 ± 0.01	0.96 ± 0.01	105.07 ± 0.59	7.45 ± 0.00
**D90H10**								
10	22.3 ± 0.2	2643 ± 50	942,497 ± 10	0.0004 ± 8 × 10^−6^	0.56 ± 0.01	0.98 ± 0.01	132.70 ± 0.72	8.39 ± 0.00
20	18.8 ± 0.2	1897 ± 6	1,311,499 ± 7	0.0003 ± 9 × 10^−7^	0.54 ± 0.01	0.99 ± 0.00	121.41 ± 0.49	9.06 ± 0.00
30	11.4 ± 0.4	1880 ± 10	1,325,515 ± 6	0.0003 ± 2 × 10^−6^	0.69 ± 0.01	0.99 ± 0.00	102.93 ± 0.68	18.74 ± 0.65
40	12.3 ± 0.2	7050 ± 50	354,106 ± 2	0.0011 ± 8 × 10^−6^	0.45 ± 0.01	1.00 ± 0.00	103.89 ± 0.38	10.82 ± 0.00
50	12.0 ± 0.2	9160 ± 20	270,073 ± 90	0.0014 ± 2 × 10^−6^	0.69 ± 0.01	1.00 ± 0.00	107.59 ± 0.80	8.77 ± 0.00

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
