# Peer review of "Curing Behavior, Rheological, and Thermal Properties of DGEBA Modified with Synthesized BPA/PEG Hyperbranched Epoxy after Their Photo-Initiated Cationic Polymerization"

_polymers, 2020, doi:10.3390/polym12102240_

Round 1
Reviewer 1 Report
In my opinion, this manuscript addresses an interesting and important subject (the curing behaviour of epoxy resins with different degrees of branching), but requires considerable revision before it could be accepted for publication.
Some issues amount to merely typographical or grammatical errors; however, others are more important (possibly requiring further work).
1) The interpretation of what is happening during curing is based on the rheology and thermal behaviour (shown in Fig. 5 and supplemantary Fig. S1). I could find no indication of how reproducible any of these data were - or even whether they were single measurements or based on multiple experiments. It is essential to provide an indication of how reproducible and representative these data are, please - otherwise, attempting to draw inferences on the differences between the systems is meaningless.
2) L134-137: The authors should provide further details regarding the photo-DSC measurements of curing, please. It is stated that the specimens were placed in aluminium pans; were these left uncovered or were transparent lids used, please?
3) The authors should state if the power output of the UV illumination was calibrated, please.
4) L149-150: What was the basis for choosing the conditions for the rheological observations of curing, please? In particular, were any preliminary measurements made to check whether a frequency of 10 Hz, at 0.1 % strain was suitable, please?
5) I cannot see that Fig. 3 provides any additional insight over the information already presented in the text. It would be better to remove the present Fig. 3 and add some representative rheology data.
Arguably, the rheology is as important as the DSC data, as it provides the basis for estimating molecular weight between cross-links.
6) L158: The authors state the Cu K-alpha wavelength as 0.158 nm; I think it is more like 0.154 nm. The authors should check, please.
7) L196: I do not understand why the authors state that 'at the gel point, there are no entanglement effects'. I can see no reason why entanglements should disappear as the material cross-links into a gel. Do the authors mean that the entanglement effects become negligible compared with the effects of covalent cross-links? The authors should explain, please.
8) L208-209: The explanation of the gelation time is unclear. What does 'all monomers approaching maximum molecular weight' mean? Could the authors express themselves more clearly, please.
9) L211 and L236: The meaning of the expression 'high activated photoinitiator' is unclear. Do the authors mean 'high concentration of activated photoinitiator'? Please clarify.
10) L217-217: The expression 'more probability during the reaction' is unclear. I suspect the authors mean 'more probability of reaction'. Please clarify.
11) L264-265: Some of the data appears to be presented to an improbably high degree of precision in Table 3. Was it really possible to measure the gel time to 0.1 s, or the Tg and Rg to 2 decimal places? The authors should provide indications of the uncertainty limits in order to justify the precission implied - or modify the way the results are presented to be more realistic.
12) It is difficult to identify the values of Tg in Fig. 9; it would help to indicate the positions (e.g. by arrows), please.
There also appear to be systemmatic variations in the magnitude of heat capacity changes at the Tg, particularly for the D9H10 system. This may be related to the number of additional vibration modes that become active above Tg. Can the authors comment, please?
13) Various grammatical or typographical errors:
L30: The meaning of the expression 'curable ambient temperature' is not clear. Do the authors mean 'curavble at ambient temperature'? The authors should check, please.
L8: The expression 'having a low viscous liquid' is incorrect. I suspect the authors mean 'presenting as a low viscosity liquid'. The authors should check, please.
L166 and Table S2: Where a line crosses the y-axis is called the 'intercept'.
Reviewer 2 Report
In my opinion the paper entitled “Curing Behavior, Rheological and Thermal Properties of DGEBA Modified with Synthesized BPA/PEG Hyperbranched Epoxy After Their Photo-initiated Cationic Polymerization” send to Polymers for revision have to be rejected without possibility of resubmission.
In general, the article is not well-written and there are many grammatical errors. The reported work is only a methodologic study with great errors and without much interest for other researchers. I am not convinced that the authors really know what they are doing. The paper does not have enough added value to be accepted in a high impact journal like Polymers.
Comments to the introduction:
- The authors should read what they have really written. For example: “The reaction activated by a photoinitiator is cleaved by UV light” and the reaction is evidently not cleaved but the photoinitiator molecule. Another is: “hyperbranched polymers are easy to synthesize, having a low viscous liquid, high….” I suppose they refers to the low viscosity, typical of dendritic structures.
- It is written “Crivello, a photopolymerization expert....” This sentence is quite offensive. He was not only “a photopolymerization expert” but he developed of a new class of protic acid photogenerators, also known as ‘Crivello salts,’ for inducing cationic polymerization of epoxy resins, which opened the door for the first wave of additive manufacturing systems and he was the best expert in this field.
- The explanation of AM and ACE mechanisms is confusing. The authors are explaining on the AM mechanism, but the previous sentence is about ACE mechanism.
Experimental.
- Report the methods you use to determine the epoxy equivalent of the HBP synthesized and some data extracted from the structural characterization.
- In part 2.7 Semi empirical equation for describing the UV curing reactions, the authors follow an article published by Andrzejewska without noticing that this article is for radical polymerization and the system used in their study is a cationic polymerization. The termination processes are not the same since terminal cations cannot interact among them as radicals do and there will be never a second order termination process.
- Why the authors use an isothermal temperature of 80 °C during photoirradiation? The advantages of photoirradiation are mainly related to the energy saving. In this case, it seems that this high temperature is needed for eliminating vitrification, but it is not proved. In fact, glycidylic resins are not very reactive in front of this type of UV- initiated cationic systems and cycloaliphatic are the best ones. The formulation selected, with HBP does not lead to any additional advantage and it seems clear that the only advantage is that the authors have previously prepared this material.
Results and discussion
- The fitting of the experimental point to linear relationship for the curing of formulation D90H10 cannot be considered. There is any type of linear correlation.
- There is no evidence that ACE mechanism predominates during UV reaction and AM mechanism predominates in the dark curing.
- I am not expert in the calculation of radius of gyration, but as an expert in epoxy thermosets I have never seen those calculations in this field, and I cannot understand their significance. SAXS determine regular distances and in the case of curing the reaction is random and I cannot suppose any type of ordering. I have calculated radius of gyration by light scattering in solution, but I cannot see the sense in networks.
- There is no optimization of the curing conditions to get a purely UV-photocuring
- I have serious doubts about the behaviour of the mixture D90H10 at 30 mW/cm2. How many times have been repeated to ascertain these values? It follows any trend.
Conclusions
- The authors state that second and primary radical terminations eliminate dangling chains in the network, which does not make sense, since the polymerization mechanism is cationic.
- It is written that “After UV irradiation in the DSC thermogram, the AM mechanism caused a second exothermic peak”. There is no prove that only AM mechanism, and no ACE, occurs during this thermal curing process.
- The authors seems to be quite surprised in the last sentence of the conclusions “It was significant that the glass transition temperature for both the DGEBA and D90H10 systems depended on the reaction mechanism, the structure of resins, and UV intensity” What they are expecting?
- Is there any advantage to add the HBP to the curing formulation?
Reviewer 3 Report
The authors reported the preparation and characterization of materials based on diglycidyl ether of
bisphenol A (DGEBA) modified with bisphenol A (BPA)/polyethylene glycol (PEG) hyperbranched epoxy resin. They presented the curing behavior, rheological and thermal properties and presented the obtained data.
However please consider some comments/ observations which might improve the quality of the manuscript:
- Abstract: the authors presented within the abstract shortly about the study and included too many details on the characterization method which is not necessary. The authors should reformulate the abstract in the way to be clear the aim of study, the methods and characterization methods and shortly the outcome obtained.
- Introduction: The authors presented within this section information on the mechanism of photopolymerization and some reactions performed on hyperbranched polymers. However, there is no clear state-of-art of the topic the authors presented. The reformulation of the introduction is needed. The information from this part should be presented in such a way that the motivation of this study to be clearly understood. Additionally, the introduction part should end up with the presentation of the purpose of the study highlighting the novelty character.
- The authors presented the synthesis procedure with the section materials and methods but no proof that the synthesis took place was inserted e.g. FTİR or/ NMR or other quantification method.
- The authors mentioned about ‘higher cross link density’ line 252 but no mention on how this crosslink density was included. The authors are requested to provide details.
- Conclusion part does not contain the main outcome of the study but some experimental observation instead. It will be interesting to find out how the outcome from three methods are matching each other and for which properties these materials are suitable for.
Reviewer 4 Report
The manuscript reportes the "Curing behavior, rheological and thermal properties of DGEBA modified with synthesized BP/PEG hyperbranched epoxy after their photo-initiated cationic polymerization", after the review following are some comments for authors:
- The tittle is quite interesting, however some details about how the manuscript was structured must be corrected, for instance, the supplementary file (SF) must be in a separate file, and the citation of data from supplementary file must be more specific in main manuscript, due the results discussion is centered in SF information.
- It is recommended that change the way which is referred DSC analysis, due in the way that is cited (photo-DSC), can be confuse for reader, also for photo-rheometer, both kind of analysis were carried out after the photo initiation, so must be DSC after photo irradiation to avoid confusion.
- For rheological analysis, how were stablished the frequency and the strain? a frequency and strain sweep was carried out or these conditions were taken from previous studies?
- It would be interesting to present the DSC curve for determination of curing conditions.
- For Tg value reported in table 3, only with low UV intensity was observed a significative change, how can be related this with structure of polymer? Literature indicates that high cross linked structure can be associated with increasing of Tg value, so in this case what happened? (details related with figure 4 and 7).
- In fig 9 ad it is not clear enough the identification of Tg, please indicate with an arrow in thermogram and for low irradiation of D90H10.
- References must be after conflict of interest and reported according with authors guidelines
Round 2
Reviewer 1 Report
I thank the authors for addressing my previous comments. However, I believe there are still considerable issues - including some new ones that appear to have occurred during a first revision. These issues must be addressed before the manuscript could be accepted for publication.
The data in table 2 is quoted to surprisingly high precision. E.g. Mc is quoted to 6 or 7 figures; Tg and Rg are quoted to 2 decimal places. The authors should also show the experimental uncertainties (and the number of repeats measured to obtain that data), in order to justify the levels of precision indicated.
The same criticism can be made of the data in the supplementary tables S1 and S2.
The 'number of cross-links per molecule' quoted also seem surprisingly small, while the molecular weight between cross-links seems very high Are they correct? Using the data given in the manuscript, I got considerably different values for these. The authors must check.
In view of what I suspect to be considerable errors in calculating the results given in Table 2, I suggest the authors should check that their conclusions are still justified.
L339-341: Actually, in the example given, the Tg should be above the operating temperature range of the HDD. (Given a working range of 25-80°C, Tg = -20°C would be outside the working range, but the resulting rubbery material may not be useful.)
L376: '...Zimm plot (1/I vs q2), as depicted in Fig. 10...' The Zimm plot is listed as Fig. 8 (not Fig. 10) - but, now, there seems to be two fig 8's.
Minor issues:
L96-97: The expression: 'Generally, the melting point is often in the materials using a toughener, such as PEG, added by commonly blending' does not make sense. Please check and correct.
L190: The sample-to-detector distance was quoted as 700 nm. (I suspect that should be 700 mm.) Please check.
L228, L240 and elsewhere: I could not find Table 3. I think this should be Table 2.
L232: Should that be Figs. 4a and 4b? It looks like this mistake has propagated through other Figure numbers (e.g. L244, 303 etc.)
L335: Figure 8 seems to come before Fig. 7.
L243: '...short-chain or low Mc, as observed in Fig. 7....' It looks like a lot of the figure numbering has become messed up. Please check and correct.
Reviewer 2 Report
The authors have answered the questions raised and the comments they made are enough to change my opinion about the work.
They completely eliminated the kinetic study and the conclusions, that can be applied to radical polymerization but not in case of cationic polymerizations.
Thank you very much for the effort you have done to improve the manuscript.
Author Response
Thank you very much for your comments and suggestions.
Author Response

(The authors gave the same response as above.)

Reviewer 4 Report
After review the corrected version of manuscript, this shows a significative improve, however still there are some points that need to be corrected. For instance the way which figures are cited in text due in section 3.1 there is not order in which figures are cited, in line 232 cite fig 5a-b, after in line 236 fig 6, then in line243 fig 7 and in fig 244 fig 5c, in line 298 fig 8, in line 303 fig 5c-d and in line 308 fig 8, so it is necessary a sequence in which figures are cited in text.
Caption of DSC thermogram must be corrected, the correct caption is fig 9.
Also I recommend that images of DSC and the-meter systems must be inserted in supplementary file in the aim to understand better the concept of photo-DSC and photo-rheology.
Round 3
Reviewer 4 Report
After review the manuscript, and considering that authors take in account the recommendations and corrections proposed, I consider that manuscript can be accepted to be published